# Predicting Prognosis and Immunotherapy Response in Multiple Cancers Based on the Association of PANoptosis-Related Genes with Tumor Heterogeneity

**DOI:** 10.3390/genes14111994

**Published:** 2023-10-25

**Authors:** Yunhan Wang, Boyu Zhang, Zongying Zhang, Jia Ge, Lin Xu, Jiawei Mao, Xiaorong Zhou, Liming Mao, Qiuyun Xu, Mengmeng Sang

**Affiliations:** 1Department of Immunology, School of Medicine, Nantong University, 19 Qixiu Road, Nantong 226001, China; 2113310012@stmail.ntu.edu.cn (Y.W.); 2031110059@stmail.ntu.edu.cn (B.Z.); 2113310013@stmail.ntu.edu.cn (Z.Z.); gejia@stmail.ntu.edu.cn (J.G.); 2031110568d@stmail.ntu.edu.cn (L.X.); 2031110575d@stmail.ntu.edu.cn (J.M.); zhouxiaorong@ntu.edu.cn (X.Z.); lmmao@ntu.edu.cn (L.M.); 2Basic Medical Research Center, School of Medicine, Nantong University, Nantong 226019, China

**Keywords:** PANoptosis, pan-cancer, tumor survival, immune cell infiltration, tumor heterogeneity, drug prediction, tumor prognosis

## Abstract

PANoptosis is a newly recognized inflammatory pathway for programmed cell death (PCD). It participates in regulating the internal environment, homeostasis, and disease process in various complex ways and plays a crucial role in tumor development, but its mechanism of action is still unclear. In this study, we comprehensively analyzed the expression of 14 PANoptosis-related genes (PANRGs) in 28 types of tumors. Most PANRGs are upregulated in tumors, including Z-DNA binding protein 1 (*ZBP1*), nucleotide-binding oligomerization domain (NOD)-like receptor pyrin domain-containing 3 (*NLRP3*), caspase (*CASP*) 1, *CASP6*, *CASP8*, *PYCARD*, *FADD*, *MAP3K7*, *RNF31*, and *RBCK1*. PANRGs are highly expressed in GBM, LGG, and PAAD, while their levels in ACC are much lower than those in normal tissues. We found that both the CNV and SNV gene sets in BLCA are closely related to survival performance. Subsequently, we conducted clustering and LASSO analysis on each tumor and found that the inhibitory and the stimulating immune checkpoints positively correlate with *ZBP1*, *NLRP3*, *CASP1*, *CASP8*, and *TNFAIP3*. The immune infiltration results indicated that KIRC is associated with most infiltrating immune cells. According to the six tumor dryness indicators, PANRGs in LGG show the strongest tumor dryness but have a negative correlation with RNAss. In KIRC, LIHC, and TGCT, most PANRGs play an important role in tumor heterogeneity. Additionally, we analyzed the linear relationship between PANRGs and miRNA and found that *MAP3K7* correlates to many miRNAs in most cancers. Finally, we predicted the possible drugs for targeted therapy of the cancers. These data greatly enhance our understanding of the components of cancer and may lead to the discovery of new biomarkers for predicting immunotherapy response and improving the prognosis of cancer patients.

## 1. Introduction

Pyroptosis, apoptosis, and necroptosis are the most well-defined programmed cell death (PCD) pathways in genetics, and they participate in the homeostasis of the internal environment and diseases in various complex ways [1]. In recent years, more and more research has focused on the crosstalk and coordination between death pathways such as cell apoptosis, cell pyroptosis, and necrotic apoptosis, and the crosstalk of these three death pathways has been integrated into a new term: PANoptosis. As an inflammatory PCD pathway regulated by the PANoptosome complex, PANoptosis has the crucial characteristics of the three death pathways that cannot be explained by any of the PCD pathways alone [2,3]. Christgen et al. found that, in macrophages infected with the influenza virus, the three cell death pathways are simultaneously activated and coordinate with each other. When one pathway is inhibited, another pathway compensates [4]. That is to say, under certain conditions, crosstalk between the PCD pathways can facilitate the transition from one death mode to another [5,6,7].

Recent studies have shown that PANoptosis is closely related to the tumor immune microenvironment (TIME) and cancer treatment [8,9,10]. The PANoptosis pathway was initially proven to contain *RIPK1*, a caspase recruitment domain (ASC), nucleotide-binding oligomerization domain (NOD)-like receptor pyrin domain-containing 3 (*NLRP3*), and caspase (*CASP*) 8 [11]. The study by Zheng et al. demonstrated that *RIPK3*, *CASP6*, Z-DNA binding protein 1 (*ZBP1*), and *CASP1* play a crucial role in promoting the activation of programmed cell death pathways (PANoptosis) and in the host defense against influenza A virus (IAV) infection [12]. In subsequent studies, Karki et al. found that adenosine deaminase acts on RNA 1 (ADAR1), limiting the *ZBP1*-mediated immune response and PANoptosis to promote tumor development [13]. ADAR1 and *ZBP1* are the only two Zα-containing mammalian proteins with a domain that is closely related to innate immunity and can mediate *NLRP3* inflammasome activation, inflammation, and cell death [14]. *ZBP1* initiates the assembly of *ZBP1*-PANoptosomes, and *CASP6* interacts with *RIPK3* to enhance the interaction between *RIPK3* and *ZBP1*, thereby promoting the assembly of PANoptosomes [15]. Many studies have confirmed that promoting PANoptosis can inhibit cancer, indicating that PANoptosis may be a potential therapeutic target [16,17,18]. However, the mechanism of PANoptosis’s role in tumors is still unclear, and the functions and regulatory patterns of its components still need to be studied.

The role of some PANoptosis-related genes (PANRGs) in some cancers has been revealed in previous studies. For example, Liu et al. found that *CASP8* polymorphisms contribute to the prognosis of advanced LUAD patients after platinum chemotherapy [19]. Rodrigues et al. showed that coordinate loss of *MAP3K7* and CHD1 promotes aggressive PRAD [20]. Liu et al., found that inhibiting *NLRP3* inflammatory body can effectively inhibit pancreatic cancer [21]. However, the connection between PANoptosis as a whole and anticancer immunity is still unclear. In addition, the function of PANRGs and their association with various tumors have not been fully explored.

In this study, we selected a set of genes that are involved in regulating the process of PANoptosis, and the association between these PANRGs and multiple cancers was studied from various dimensions, which will aid future studies in exploring the potential mechanisms. We evaluated the clinical characteristics, survival status, and prognostic value of PANRGs in 28 tumors. Further evaluation was conducted on their relationships with immune checkpoints, infiltrating immune cells, tumor stemness, tumor heterogeneity, miRNAs, and identification of potential drug therapies. These results will enhance our understanding of the mechanism of PANoptosis and aid in the design of new strategies for treating tumors.

## 2. Materials and Methods

### 2.1. Data Collection

We downloaded the gene expression data of normal tissues from the GTEx project (https://gtexportal.org/home/). The Cancer Genome Atlas (TCGA) database (http://cancergenome.nih.gov) is a comprehensive database containing multiple omics datasets for various cancers. Since there are tissue data for only 28 types of cancer in the GTEx database, we subsequently downloaded the corresponding expression profiles, clinical information, and mutation data for these 28 types of cancer from the TCGA database [22,23].

### 2.2. Data Processing

All analyses were performed using R software (v 4.2.3). For the TCGA data, differential expression analysis was performed using the Wilcoxon rank sum and signed rank tests to compare the differential expression of genes of interest. All correlation analyses were conducted using the cor.test function in R.

### 2.3. Immunohistochemical Analysis of PANRG Expression in Multiple Cancers

Due to the high specificity of antigen and antibody binding, immunohistochemistry (IHC) can reveal the relative distribution and abundance of proteins. The Human Protein Atlas (HPA, https://www.proteinatlas.org) is an online database that contains a large amount of protein expression data from IHC tissue microarrays in normal and cancer tissues [24]. Through the HPA, we can observe the expression of PANRGs in glioma, KIRC, PAAD, STAD, and TGCT more intuitively.

### 2.4. Survival Analysis of Copy Number Variants (CNVs), Single-Nucleotide Variations (SNVs), and Methylation Levels

The Cancer Variant Nexus (CVN) database was used to obtain somatic mutation data for multiple cancers. All missense mutations, nonsense mutations, insertions and deletions, splice site variants, and frameshift mutations were included in our analysis. We used the R package “maftools” (v 2.16.0) to extract information about the variant type, genomic coordinates, and gene symbols. We downloaded survival information for multiple cancers from the TCGA database, including overall survival (OS), progression-free survival (PFS), disease-free interval (DFI), and disease-free survival (DSS), to evaluate the relationship between survival in multiple cancers and PANAG CNVs, SNVs, and methylation levels.

### 2.5. Molecular Clustering of Multiple Cancers

Consensus clustering is a widely used method in unsupervised learning that aims to determine the optimal number of clusters by minimizing the variability between different clustering algorithms. Cluster analysis was performed using ConsensusClusterPlus [25], using agglomerative PAM clustering with 1-Pearson correlation distances and 80% resampling of the samples for 10 repetitions. The optimal number of clusters was determined using the empirical cumulative distribution function plot.

### 2.6. PANRG Risk Model Construction

To further refine our analysis, we applied the “glmnet” (v 4.1-7) (33) and “survival” (v 3.5-5) packages for LASSO regression to screen for candidate PANRGs. The LASSO method is a widely used approach in high-dimensional data analysis that can select relevant features and exclude irrelevant ones by shrinking their coefficients towards zero. In this study, we used the R software package glmnet to integrate survival time, survival status, and gene expression data, and we used the LASSO box method for regression analysis. In addition, we also set a 10-fold cross-validation to obtain the optimal model. Subsequently, we used the R software package maxstat (maximally selected rank statistics with several *p*-value approximations, v 0.7-25) to calculate the optimal cutoff value for the Risk Score and set the minimum group sample size to be greater than 25% and the maximum group sample size to be less than 75%. Finally, the optimal cutoff value was obtained. Based on this, patients were divided into high and low groups, and further analysis of the pre- and post-diagnosis differences between the two groups was performed using the R software package survival’s survival function. We evaluated the significance of prognostic differences between different groups of samples using the log rank test method, and we ultimately observed significant prognostic differences (*p* = 4.7 × 10^−7^). In addition, based on this model, we conducted subsequent analyses, such as a survival analysis and drug predictions, to gain further insights into the underlying mechanisms of multiple cancers.

### 2.7. Immune Infiltration Analysis

To elucidate the differences in immune infiltration between different PANRG clusters in multiple cancers, we used the “deconvo_epic” (v 1.1.0) algorithm of the IOBR package (v 0.99.9) (https://www.ncbi.nlm.nih.gov/pmc/articles/PMC8283787/ (accessed on 22 October 2023)) to evaluate the immune cell scores.

### 2.8. The Stemness Features

Based on Malta et al.’s research, we collected DNAss (DNA methylation-based), EREG-METHss (epigenetically regulated DNA methylation-based), DMPss (differentially methylated probe-based), ENHss (enhancer element/DNA methylation-based), RNAss (RNA expression-based), and EREG.EXPss (epigenetically regulated RNA expression-based) tumor dryness scores calculated using the mRNA expression levels and methylation signature for each tumor [26]. We calculated the correlation between the stemness features and PANRG expression.

### 2.9. Tumor Heterogeneity

We used the tmb function of maftools to calculate the tumor mutation burden (TMB) score for each tumor and inferHeterogeneith to calculate the MATH score (mutant-allele tumor heterogeneity). We calculated the microsatellite instability (MSI) score based on Bonneville et al.’s study [27], and we calculated the neoantigen (NEO) score, purity score, ploidy score, homologous recombination deficiency (HRD) score, and loss of heterozygosity (LOH) score based on Thorsson et al.’s study [28]. We calculated the correlation between the tumor heterogeneity and PANRG expression levels.

### 2.10. Analysis of Drug Sensitivity of Cancers Expressing PANRGs

We obtained drug sensitivity data from The Genomics of Drug Sensitivity in Cancer (GDSC) database (https://www.cancerrxgene.org/). The R package “oncoPredict” (v 0.2) was used to download the IC50 values of each drug. Subsequently, we performed a correlation analysis between drug sensitivity and the expression of PANRGs. By analyzing the relationship between drug sensitivity and PANRG expression in different patient groups and subtypes, we aimed to identify potential targets for future treatments and to provide insights into personalized therapeutic strategies for cancer patients.

### 2.11. Statistical Analysis

All statistical analyses in this study were conducted using R software (v4.2.3; The R Foundation for Statistical Computing), which is a widely used open-source platform for statistical computing and graphics. To ensure the robustness and reliability of our findings, we utilized two-tailed statistical tests for all our analyses. A significance threshold of *p* < 0.05 was applied to determine statistical significance, as recommended in previous studies.

## 3. Results

### 3.1. Prognostic Impact of the PANRGs in Different Cancer Types

To determine whether PANoptosis-related genes have prognostic value in multiple cancers, we first defined a set of molecules involved in the regulation of PANoptosis and obtained RNA-seq data for 28 different cancer types from the TCGA database (Abbreviation). Here, the fourteen PANoptosis-related genes, namely, *NLRP3*, *CASP8*, *CASP6*, *CASP1*, *RIPK1*, *MAP3K7*, *TNFAIP3*, *FADD*, *RIPK3*, *RNF31*, *PYCARD*, *PSTPIP2*, *RBCK1*, and *ZBP1*, were identified as components of the PANoptosome according to recent studies (Figure 1A). Most of these genes are widely expressed in a variety of cancers and might exert their roles in regulating PANoptosis via protein–protein interactions (PPIs) (Figure 1B). At the same time, we also analyzed the protein expression levels of these genes in various tissues and determined their distribution in the body (Figure 1C). We then analyzed the expression of these genes across the 28 different cancer types using data from the TCGA and GTEx databases. The results showed that the expression of most of the PANRGs were upregulated in 10 cancers, namely, GBM, LGG, ESCA, KIRP, COAD, STAD, HNSC, KIRC, PAAD, and TGCT, and downregulated in 8 cancers, namely, BRCA, LUAD, PRAD, LUSC, THCA, UCS, ACC, and KICH. The upregulated genes were *ZBP1*, *NLRP3*, *CASP1*, *CASP6*, *CASP8*, *PYCARD*, *FADD*, *MAP3K7*, *RNF31*, and *RBCK1* (Figure 1D).

Based on the differential expression of PANRGs in tumors and normal tissues, we analyzed the correlation between PANRG expression and patient survival using univariate Cox regression analysis, finding that the gene *ZBP1* was a risk factor for LGG, KIRC, PAAD, and LAML. *NLRP3* was a risk factor for LAML. *RIPK1* was a risk factor for LGG, LAML, and ACC. *RIPK3* was a risk factor for LGG, PAAD, and LAML. *CASP1* was a risk factor for LGG, PAAD, TGCT, and LAML. *CASP6* was a risk factor for LGG, LUAD, LIHC, PAAD, and KICH. *CASP8* was a risk factor for LGG, LIHC, and PAAD. *PYCARD* was a risk factor for LGG, KIRC, PAAD, and LAML. *FADD* was a risk factor for LGG, CESC, LUAD, PRAD, HNSC, LUSC, LIHC, PAAD, and LAML. *MAP3K7* was a risk factor for LGG and LIHC. *TNFAIP3* was a risk factor for LGG, CESC, GBM, and ACC. *RNF31* was a risk factor for READ, LAML, and ACC. *RBCK1* was a risk factor for LGG, LUAD, KIRP, HNSC, KIRC, LUSC, LIHC, and LAML. *PSTPIP2* was a risk factor for LGG, KIRP, LAML, and KICH (Figure 2). In summary, *FADD* and *RBCK1* are risk factors capable of influencing the largest number of cancers; *CASP1*, *CASP6*, and *FADD* are highly expressed in tumors; and LGG is the most susceptible cancer to therapies targeting PANRGs.

Through the IHC data in the HPA database, we verified that the expression of most PANRGs in glioma, KIRC, PAAD, and THCT was higher than that in normal tissues (Appendix A). We focused on examining the expression of one of the PANRGs, *CASP1*, and found that its expression was increased in a number of cancers (Appendix A). This conclusion was also confirmed by the IHC data (Appendix A). Together, these data demonstrated that most of the PANRGs were aberrantly expressed in multiple cancers, indicating that some PANRGs might be risk factors for particular cancers.

### 3.2. The Relationship between PANRGs and Clinicopathological Features

To determine the impacts of PANRGs on clinical indicators in each cancer, we analyzed the degree to which these genes play a role in tumors based on age, stage, grade, and gender. We divided the age group into two subgroups based on an age cut-off of 60 years and found that KIRP is the most highly correlated with age and expresses most of the PANRGs (Figure 3A). Based on the cancer stage, OV, PAAD, and THCA were significantly correlated with stages (Figure 3B). PRAD is regulated by most of the genes in the N stage, while the *PSTPIP2* gene can regulate most types of the tumors (Figure 3C). In the M stage, the expression of *PYCARD* and *RBCK1* was correlated with KIRC with the highest significance level (Figure 3D). In the T stage, *RIPK1*, *RIPK3*, and *CASP8* were the most highly expressed genes in cancers, while *ZBP1*, *PYCARD*, and *RBCK1* showed the highest correlation with cancers (Figure 3E). Next, we analyzed the impact of the genes on tumors in different genders and found that *NLRP3* had a relatively broad impact on tumors. The number of differentially expressed genes in LUAD and LUSC was greater than that of other cancers (Figure 3F). Finally, the expression of most PANRGs exhibited significant differences in the different grades of LGG (Figure 3G). In summary, these findings expand our understanding of the associations between various clinical features of cancers and the expression of PANRGs (the multiple gene comparison maps for the various tumors can be found in Appendix A).

### 3.3. The Association between PANRGs in Multiple Cancers

To further explore the relationship of PANRGs in different cancers, we analyzed the expression levels of PANRGs in 28 types of tumors (Figure 4). In LGG, LIHC, and OV, the correlation between genes was almost always positive. However, in CHOL and TGCT, the genes exhibited a strong negative correlation. Overall, most of the PANRGs in tumors were positively correlated with one another. The synergistic expression or activation of the PANRGs may indicate the occurrence of some other types of PCD, which may concomitantly promote the cancer progression. In comparison, the small number of negatively correlated genes observed in our analysis may indicate a possible dominance of a specific type of cell death in tumors, a phenomenon that also worth further investigating.

### 3.4. Genetic Alterations in PANRGs Affect Survival

Genetic changes are important factors for tumorigenesis. Thus, we analyzed the effects of CNVs, SNVs, and methylation of PANRGs on the survival of cancer patients. Figure 4A shows the relationship between CNVs in the PANoptosis gene set and the survival of cancer patients. A comprehensive analysis of the four dimensions of OS, PFS, DSS, and DFI found that the gene set CNVs were most closely related to the survival of BLCA patients, followed by LGG and LIHC patients. KIRP and THCA also showed strong associations with gene set CNVs in PFS and DSS (Figure 5A). By analyzing the differential survival rates of patients bearing wild-type or mutant gene sets, it was found that genetic variations had the greatest influence on the survival rate of BLCA (Figure 5B). Further analysis of the impact of gene set SNVs in tumors on OS revealed that when a *CASP8* SNV was present in COAD, wild-type individuals had a longer lifespan. On the contrary, the UCEC patients bearing *CASP8* mutations exhibited a longer lifespan. This is consistent with the overall survival rate of patients bearing mutant *NLRP3* SNVs being higher in UCEC and BLCA. In STAD, wild-type *PSTPIP2* SNVs confer a higher survival rate for patients. In BRCA, the survival rate of patients with wild-type *RIPK3* SNVs was much higher than that of individuals bearing mutant SNVs (Figure 5C–H). Additionally, we conducted methylation analysis to investigate the impact of methylation levels on the survival of patients bearing various tumors. We found that the survival rate of LGG patients was the most highly correlated with methylation, followed by that of ACC patients (Figure 5I). These data indicated that the presence of CNVs, SNVs, and high methylation levels in PANRGs is closely related to the survival of cancer patients.

### 3.5. Clustering Analysis

To group the tumor samples based on the expression profiles of PANRGs, we performed unsupervised consensus clustering separately for each cancer type (Figure 6A). Moreover, in each cancer, we also analyzed the differences between these groupings. The results showed that the genes showed grouping variability in most tumors, with *PYCARD* and *RBCK1* being the most significant. However, almost all the genes, except *RBCK1*, in READ did not show differential expression between groups (Figure 6B and Appendix A).

### 3.6. Risk Factors Predicted by LASSO-Cox Analysis

We then performed LASSO-Cox analysis to identify genes that play key roles in the regulation of various tumors (Figure 7). The minimum size of the sample group was set to be greater than 25%, and the maximum size was set to be less than 75%. The optimal cutoff value for the Risk Score was obtained, based on the groups that the patients were classified into, to compare the differences in survival between the two groups (Figure 8A). All 16 types of cancers showed consistent performance, and the survival rate of the low-Risk Score group was higher than that of the high-Risk Score group (Figure 8A). Based on the above results, we ultimately chose the LASSO-Cox regression model to predict the recurrence risk of multiple cancers patients after treatment. To facilitate clinical practice, we converted complex mathematical models into nomograms. Based on the nomograms, we can predict recurrence after 1 year, 3 years, and 5 years (Figure 8B). Figure 8C shows the differential expression of genes in various tumors by high or low group (Figure 8C and Appendix A). (LASSO analysis and calculation data can be found in Appendix A.)

### 3.7. Correlation between PANRGs and Immune Checkpoint Genes

To explore the immune function of PANoptosis-related genes in cancer, the correlation between the expression of the PANRGs and immune checkpoint suppressor genes in 28 cancer types was first calculated (Appendix A). *ZBP1*, *NLRP3*, *CASP1*, *CASP8*, and *TNFAIP3* levels were positively correlated with the levels of almost all immune checkpoint suppressor genes in all tumors (Appendix A). Based on the clustering analysis of immune checkpoint inhibition in various tumors, we found that high expression of these immune checkpoint genes was highly correlated with THCA (Figure 9A and Appendix A). When the patients were divided into high and low groups, the difference in LIHC was the most significant. Except for IL-13, the expression of all other inhibitory immune checkpoint genes showed significant differences between groups (Figure 9B and Appendix A). At the same time, we also analyzed the correlation between the PANRGs and the stimulatory immune checkpoint genes (Appendix A). Similarly, *ZBP1*, *NLRP3*, *CASP1*, *CASP8*, and *TNFAIP3* were positively correlated with most stimulatory immune checkpoint genes in the 28 types of cancers (Appendix A). In the cluster analysis of tumors, stimulatory immune checkpoint genes were most strongly correlated with THCA (Figure 9C and Appendix A), while in the LASSO analysis, these genes were most strongly correlated with TGCT (Figure 9D and Appendix A).

### 3.8. Correlation between PANRGs with Tumor Immune Cell Infiltration

To clarify the relationship between the PANoptosis score and immune cell infiltration in the tumor microenvironment, we conducted a correlation analysis between the different cancers. Based on gene expression, the infiltration scores of 22 immune cells were evaluated for each patient in each tumor. The PANRGs were positively correlated with the infiltration levels of activated CD4^+^ memory T cells and M1 macrophages in ACC, CESC, HNSC, LGG, LIHC, OV, PAAD, PCPG, READ, SKCM, and UCS. Plasma cells showed a negative correlation with BLCA, BRCA, HNSC, LUAD, LUSC, OV, PAAD, STAD, TGCT, and UCEC (Appendix A). Clustering analysis of immune cells showed significant differences in levels of CD8^+^ T cells, CD4^+^ resting memory T cells, follicular T helper cells, regulatory T cells (Tregs), and M1 macrophages in tumors (Figure 10A and Appendix A). LASSO analysis showed that CD4^+^ resting memory T cells, follicular T helper cells, M1 macrophages, and M2 macrophages exhibited significant differences in tumors with high and low expression of PANRGs (Figure 10B and Appendix A).

### 3.9. The Relationship between PANRGs and Tumor Stemness

We calculated six tumor stemness indices using mRNA expression and methylation signature data from previous studies, namely, RNAs, ENHss, EREG.EXPss, DNAss, EREG-METHss, and DMPss [26]. Most PANRGs were positively correlated with tumor stemness in BLCA, CESC, COAD, GBM, HNSC, KIRC, LUAD, LIHC, and UCEC (Figure 11). In the analyzed tumors, PANRGs in LGG showed a strong positive correlation with the tumor dryness indicators but showed a strong negative correlation with RNAss. Through cluster analysis of tumor stemness, BRCA, PRAD, STAD, and TGCT were closely related to tumor stemness, with all tumor stemness clusters in STAD being different (Appendix A). The LASSO analysis showed that the difference between EREG.EXPss and RNAss had the highest significance level (Appendix A). Based on the clustering analysis and LASSO prognostic analysis, PANRGs are associated with different levels of tumor stemness in different cancers.

### 3.10. The Association between PANRGs and Tumor Heterogeneity

To further evaluate the potential effects of the PANRGs on tumor heterogeneity, we analyzed the correlations between the expression levels of PANRGs and seven indices that reflect the heterogenous features of cancers, namely, TMB, MATH, MSI, purity, ploidy, HRD, and LOH. Based the clustering analysis results, the expression of PANRGs was found to be significantly correlated with heterogeneity indices, including purity, HRD, and LOH, in approximately half of the cancers, while PANRGs were not significantly correlated with TMB and NEO in almost all cancers (Figure 12A and Appendix A). The LASSO analysis results showed a difference between the high and low risk groups, mainly in purity, HRD, and LOH, while almost no differences were observed in TMB, MSI, and NEO (Figure 12B and Appendix A). These data reveal the relationship between PANRGs and tumor heterogeneity in cancer.

### 3.11. The Relationship between PANRGs and Regulatory miRNA Expression in Cancer

To investigate whether PANRGs are involved in the regulatory role of miRNAs in various cancers, we analyzed the correlation between the expression levels of PANRGs and various miRNAs. We found that the expression of *MAP3K7* showed positive or negative correlations with the expression of 18 miRNAs (Figure 13). In comparison, the number of miRNAs correlated with other PANRGs was much smaller than that of *MAP3K7* (Figure 13). This result suggests that PANRGs may be correlated with miRNAs to varying degrees.

### 3.12. Drug Predictions Based on Functions of PANRGs May Improve the Treatment of Cancer

Based on the above analysis, we then asked if PANRGs are potential targets for drug development to treat cancer. To this end, we employed oncoPredict to predict the correlations of PANRGs to existing drugs. The top 10 predicted drugs of each cancer based on IC50 were collected and are illustrated in Figure 14. Overall, the predicted drugs had positive or negative correlations with some of the PANRGs in each cancer. There were significant differences in the impact of some drugs between the high and low risk groups. For instance, all the drugs may have a role in regulating the expression of a particular PANRG, while only a few drugs, including Bortezomib, Vinblastine, Staurosporine, and Dinaciclib, showed different effects in high and low risk groups of ACC patients. Among these drugs, Staurosporine manifested the most significant impact on different patient groups of ACC, possibly due to its relatively lower positive correlations with PANRGs (Figure 14). Thus, these data demonstrated that drug prediction using PANoptosis-related genes may facilitate the development of new drugs and improve the treatment of cancer.

## 4. Discussion

Programmed cell death is a core regulatory mechanism of many physiological processes, such as development, homeostasis, and immune responses. As important terminal pathways for multicellular biological processes, the dysfunction of PCD pathways may lead to the development of many diseases, including cancers [29]. Apoptosis, pyroptosis, and necroptosis are the three most well-studied PCD pathways in the literature and were previously described as distinct pathways. However, growing evidence regarding the extensive crosstalk of these pathways has led to the establishment of a new PCD-associated concept, PANoptosis. It has been recognized as a new type of PCD pathway and may have the potential to overcome apoptosis resistance and activate the PCD pathway by inducing inflammatory cell death in tumors [2,30]. More and more evidence suggests that PANoptosis plays an important role in cancer treatment and may be a new therapeutic target [1,9,10]. However, the specific mode and mechanism of action of PANoptosis in various tumors are not yet understood.

Based on data from the TCGA and GTEx databases, we explored the relationship between 14 PANRGs and 28 types of cancer and found that the expression level of PANRGs in GBM, LGG, and PAAD was higher than that in normal tissues, while in ACC, the expression level of these genes was lower than that in normal tissues. Most PANRGs were upregulated in tumors, including *ZBP1*, *NLRP3*, *CASP1*, *CASP6*, *CASP8*, *PYCARD*, *FADD*, *MAP3K7*, *RNF31*, and *RBCK1*. We then conducted a survival analysis on the expression of genes in cancer, finding that *CASP1*, *CASP6*, and *FADD* are high-risk factors for the survival of cancer patients. *CASP1* plays an important role in promoting the activation of PANoptosis and host responses against infections [4,12]. Its activation also contributes to tumorigenesis via various mechanisms such as promoting the pro-tumor action of tumor-associated macrophages [31]. On this basis, we further asked if the expression of *CASP1* correlated with the progression of cancer and found that the levels of *CASP1* in 28 types of cancer were much higher than that of normal tissues, suggesting that *CASP1* may play a common pro-tumor role in multiple cancers.

To further understand the potential relationship between PANRGs and the clinical indicators in each cancer, we performed an analysis from multiple aspects such as age, stage, grade, and gender and found that the expression of PANoptosis gene sets varied in patients with different clinical indicators across various cancers. In terms of patient age, KIRP was the cancer most closely associated with PANRGs, while OV, PAAD, and THCA were closely related to PANRGs based on stage. From a gender perspective, compared to other tumors, LUAD and LUSC were more closely associated with PANRGs. In terms of tumor grade, LGG was associated with the most PANRGs. These findings suggest that PANRGs may have different expression profiles and thus affect the progression of cancers through various mechanisms. Within the 28 types of cancer, the PANRGs were generally positively correlated to one another, indicating an extensive coordinated relationship between these genes. Further investigation of the potential cooperation between PANRGs in different cancer dimensions could expand our knowledge of the role of these genes in regulating the progression of various cancer types.

The analysis of the possible genetic alterations in PANRGs is another aspect to consider in understanding the regulatory roles of these genes in cancer. We thus analyzed the impacts of CNVs, SNVs, and methylation levels of PANRGs on the survival of cancer patients. Regarding the CNV analysis, the PANoptosis gene set was found to be the most highly correlated with survival in BLCA. Further analysis showed that the SNVs in *CASP8*, *NLRP3*, *PSTPIP2*, and *RIPK3* were the most closely associated with the survival rates of tumors, indicating that SNVs caused changes that can promote tumor progression. For example, the survival rates of STAD patients bearing wild-type *PSTPIP2* and BRCA patients bearing wild-type *RIPK3* were significantly higher than those of patients with mutant genes. According to previous reports, cells with *CASP8* mutations are resistant to exogenous agents that induce cell apoptosis, which can prevent the killing of tumor cells by T cells through FasL–Fas interactions, which is one of the immune evasion mechanisms of tumors [32,33]. The *NLRP3* inflammasome is a key component of the innate immune system, and depletion of *NLRP3* in macrophages is beneficial for M1/M2b polarization [34]. *NLRP3* depletion significantly inhibited tumor growth and stage progression, and it also significantly reduced the occurrence of pancreatic ductal adenocarcinoma (PDAC) lung metastasis [34]. Methylation level was most closely related to survival in LGG and ACC. These results indicate that the genetic changes in PANRGs play a crucial role in tumor progression. Subsequent mechanistic studies may reveal the underlying regulatory molecules and associated signaling pathways for a particular cancer.

We conducted consensus clustering analysis on PANRG expression in tumors, and the CDF curve and the area under the curve indicate the optimal number of clusters. Next, we conducted LASSO-Cox regression analysis and identified some of the most significant genes in each tumor. After dividing the screened gene set into high and low risk groups, it was found that the low-risk group had a higher survival rate. Our data demonstrated that the LASSO-Cox regression model is a valuable tool for predicting the recurrence risk of cancer patients after treatment. We also converted our data to nomograms, which may serve as a useful tool to facilitate clinical practice and predict the incidence of cancer recurrence after 1 year, 3 years, and 5 years. Other nomograms predicting the risk and prognostic factors of many cancers have been produced in recent studies. For instance, Zhang et al. established a nomogram to predict the diagnosis of pancreatic cancer based on multiple risk factors using multivariable logistic regression analysis [35]. While the relative efficiency of the nomograms produced in our study and by others using different strategies for predicting the prognosis of cancers needs to be compared and validated in clinical applications in the future.

Expanding evidence suggests that the TME plays a crucial role in immune infiltration, tumor progression, and metastasis [36]. In multiple cancers, the expression of the PANRGs was closely related to most infiltrating immune cells. For instance, most PANRGs showed a positive correlation with CD4^+^-activated memory T cells and M1 macrophages and a negative correlation with plasma cells in most of the cancers. These observations are in agreement with previous reports showing increased tumor infiltration of a subgroup of CD4^+^-activated memory T cells that might be associated with improved survival for many cancers such as breast cancer [37] and oropharyngeal squamous cell carcinoma (OPSCC) [38]. The increased infiltration of M1-tumor-associated macrophages in cancers was also reported by many studies such as the one by Garrido-Martin et al., which showed that this type of macrophage might boost tissue-resident memory T cells and predict an improved survival of patients with lung cancer [39]. The results of our study provide further evidence that the infiltration of activated T cells and M1 macrophages is a common characteristic of many cancers and may thus be utilized as a biomarker for predicting the survival in many cancer types. However, the negative correlation between PANRGs and plasma cells revealed by our study was inconsistent with the findings by Patil et al., which showed that increased infiltration of plasma cells is correlated with increased survival of patients with non-small cell lung cancer (NSCLC) [40]. This discrepancy may be attributed to the variable TMEs and various types of plasma cells that may present in the TME of a particular cancer. The correlation between PANRGs and these infiltrated cells may suggest a role of these genes in regulating the proliferation or migration of immune cells to tumor tissues. Although the related mechanisms have not been clarified, there is evidence indicating that some genes that regulate PANoptosis, such as *NLRP3*, may play a role in modulating the functions of tumor-associated macrophages [41].

More and more evidence suggests that tumors originate from cancer stem cells (CSCs) [42,43,44], which can drive tumor growth [45]. In this study, six tumor stemness indices were calculated based on mRNA expression and methylation signatures. The PANRGs in LGG were the most highly correlated with tumor stemness, and almost all of them showed a strong positive correlation; however, almost all genes had a strong negative correlation with RNAss. On the contrary, LIHC was less affected by tumor stemness. The study of PANRGs and tumor stemness may aid in the future exploration of approaches to inhibit tumor cell proliferation. We also found that PANRGs have a significant impact on tumor heterogeneity in many cancers, which is consistent with previous research findings [46]. In addition, we found that PANRGs affect miRNAs to varying degrees. Based on the above analysis, we also explored the predictive ability of PANRGs for drug therapy and found that developing drugs targeting PANRGs is beneficial for cancer treatment [47].

Based on the above research, we can perform more targeted studies on the relationship and mechanism of action between PANRGs and cancer. Based on the expression level of PANRGs, we need to focus on several cancers such as GBM, LGG, and PAAD. From the perspectives of the clinical analysis, CNVs, SNVs, methylation levels, immune checkpoint status, immune cell infiltration, tumor stemness, tumor heterogeneity, and miRNA expression, compared to other genes, *ZBP1*, *NLRP3*, *CASP1*, *CASP8*, and *MAP3K7*, showed a close correlation with tumors. Thus, we can focus our research on these cancers and genes to explore new cancer treatment targets.

However, our research has some limitations. Firstly, the data obtained from the GTEx and TCGA databases exhibit heterogeneity that, to some extent, limits the accuracy of our analysis. Secondly, verifying the relationship between 14 PANRGs and 28 types of cancer is a huge workload and difficult to achieve in a short period of time. Do they play a role internally (in tumor proliferation, migration, and invasion), externally (immunosuppressive), or both? Finally, no single gene can exhibit the same association or function in 28 types of cancer, and a particular gene will have different degrees of significance in different analysis indicators of a cancer. Further research is needed on the specificity and applicability of these genes.

## 5. Conclusions

In summary, our work provides a systematic framework that identifies the correlations between PANRGs and many cancers, which may contribute to tumor progression and can be targeted to improve the prognosis of cancer patients. We constructed molecular subtypes related to PANoptosis and constructed a prognostic model, incorporating the clinical pathological characteristics, survival rate, immune checkpoint status, immune infiltration level, tumor cell stemness, tumor heterogeneity, miRNA levels, and medical treatment of cancer patients. Our research findings indicate that the levels of PANRGs have close association with tumor progression, deepening our understanding of PANRGs in cancer and providing insights for the discovery of new prognostic and immunotherapy response biomarkers.

## Figures and Tables

**Figure 1 genes-14-01994-f001:**
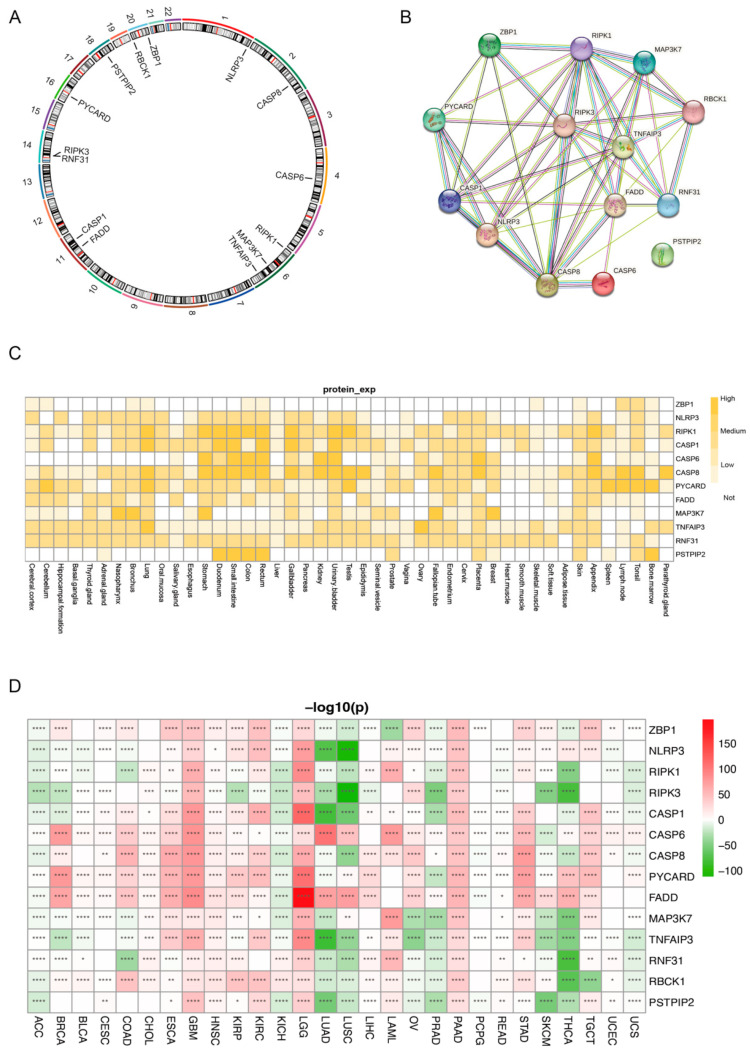
Expression levels of PANoptosis-related genes (PANRGs). (**A**) Fourteen genes related to PANoptosis. (**B**) PANRGs regulate PANoptosis through protein–protein interactions (PPIs). (**C**) Protein expression levels of PANRGs in various tissues. (**D**) Analysis of gene expression differences in various normal and cancer tissues based on The Cancer Genome Atlas (TCGA) and GTEx databases. (*: *p* < 0.05, **: *p* < 0.01, ***: *p* < 0.001, ****: *p* < 0.0001).

**Figure 2 genes-14-01994-f002:**
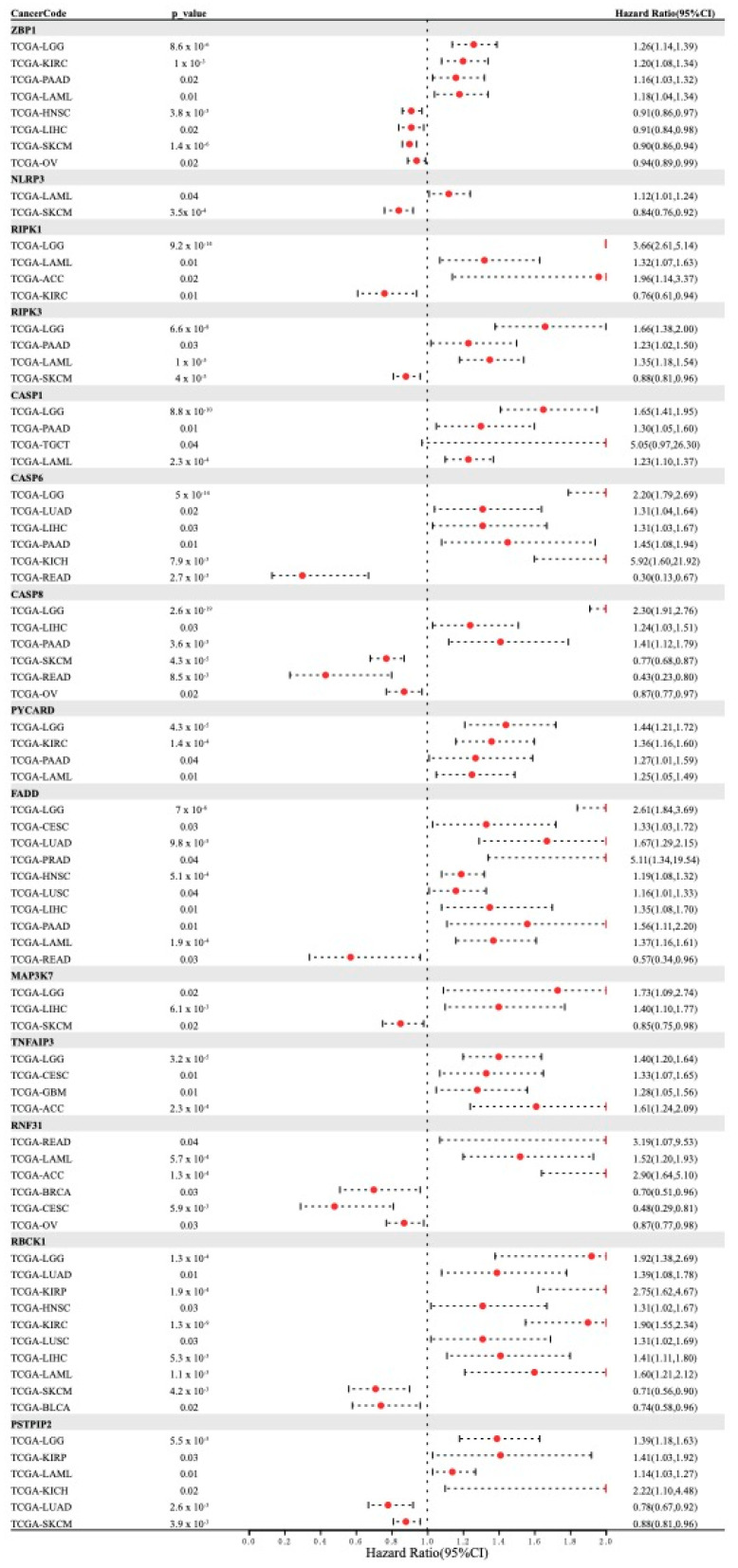
Correlation analysis between 14 PANRGs and cancer patient survival for 28 types of tumors. *p*-value less than or equal to 0.05 and hazard ratio greater than 0 indicates that the gene is a risk factor for the tumor.

**Figure 3 genes-14-01994-f003:**
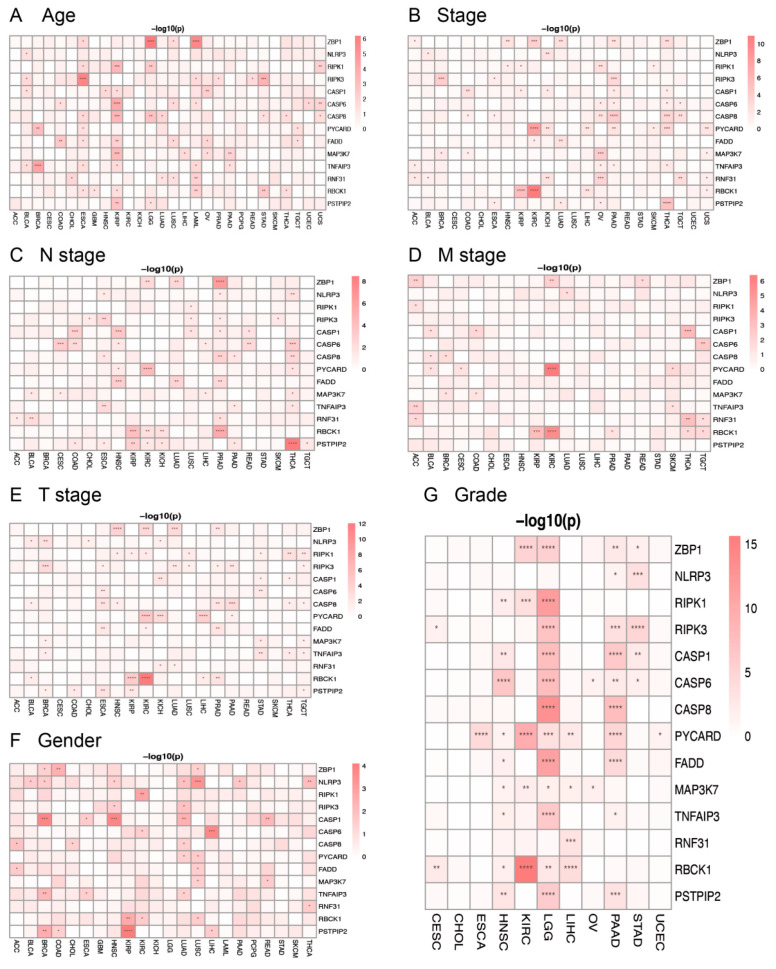
Analysis of the relationship between PANRGs and various clinical indicators of cancers: (**A**) age, (**B**) stage, (**C**) N stage, (**D**) M stage, (**E**) T stage, (**F**) gender, and (**G**) grade. (*: *p* < 0.05, **: *p* < 0.01, ***: *p* < 0.001, ****: *p* < 0.0001).

**Figure 4 genes-14-01994-f004:**
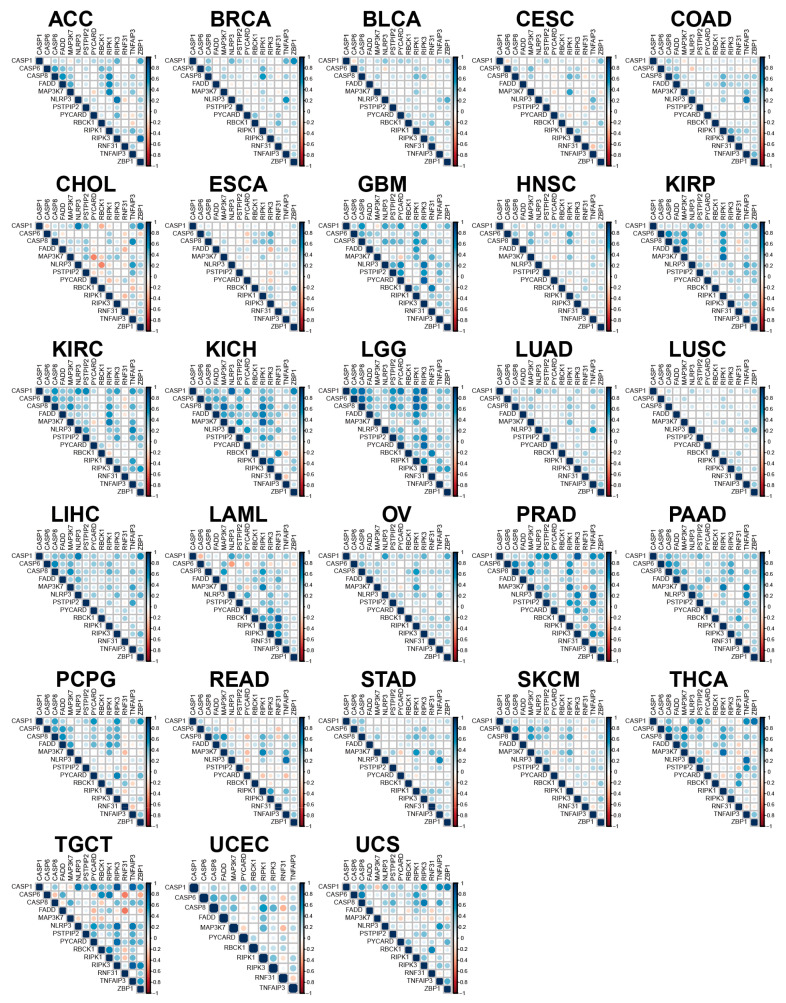
In 28 types of tumors, the interactions between 14 PANRGs were analyzed to study the correlation between the genes.

**Figure 5 genes-14-01994-f005:**
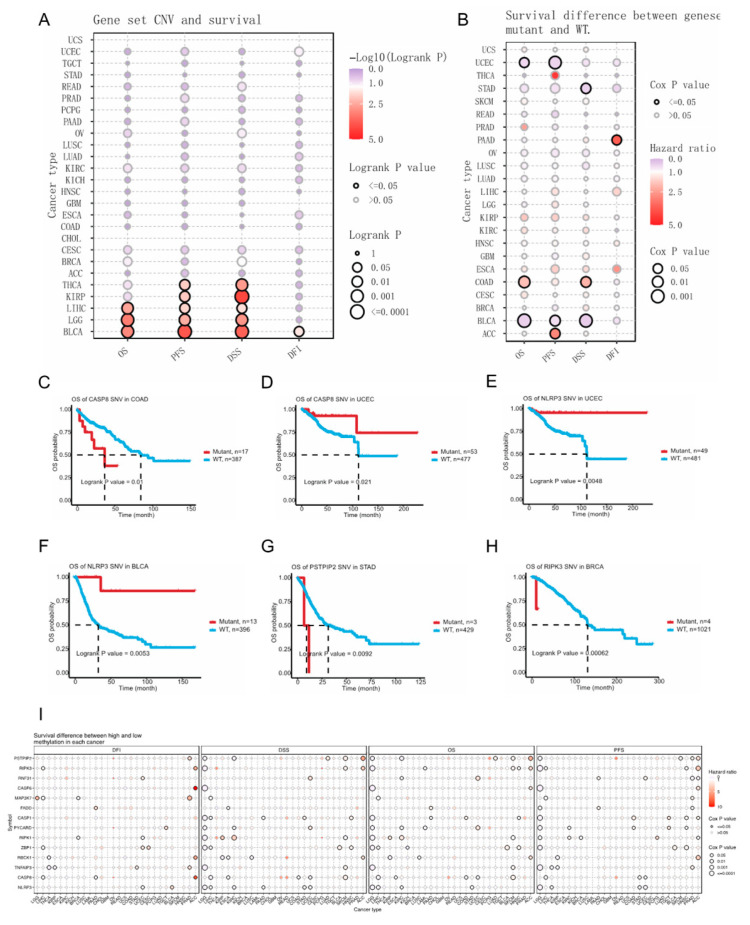
The relationship between survival and PANRG copy number variants (CNVs), single-nucleotide variations (SNVs), and methylation levels. (**A**) The relationship between CNVs in the PANoptosis gene set and the survival of tumor patients. (**B**) Analysis of survival differences between mutant and wild-type genes in various tumors. (**C**–**H**) The impact of gene set SNVs in tumors on overall survival (OS). (**I**) The impact of methylation levels on the survival of cancer patients.

**Figure 6 genes-14-01994-f006:**
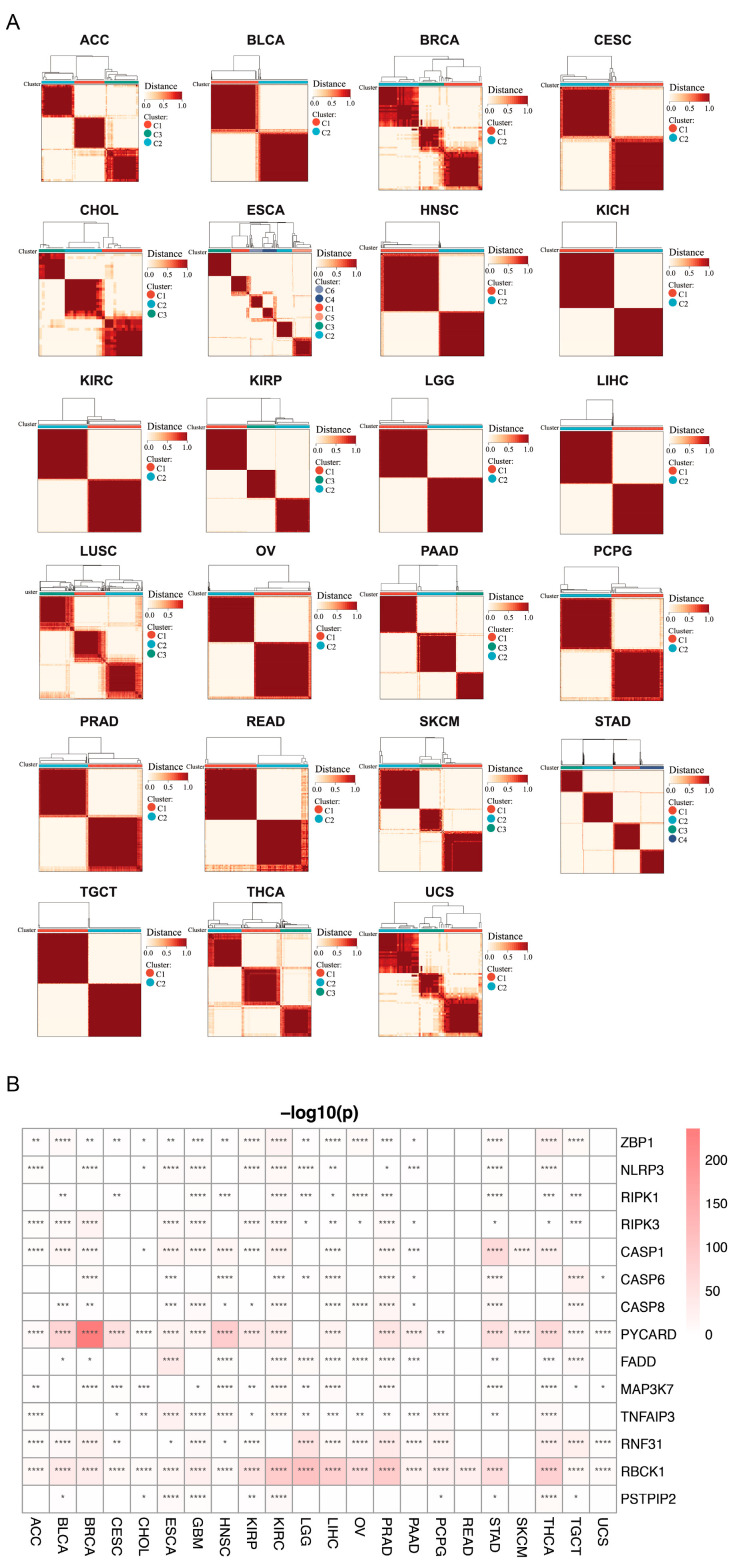
Cluster analysis of genes in various tumors. (**A**) Cluster analysis of different tumors. (**B**) The correlation between PANRGs and various tumor clusters. (*: *p* < 0.05, **: *p* < 0.01, ***: *p* < 0.001, ****: *p* < 0.0001).

**Figure 7 genes-14-01994-f007:**
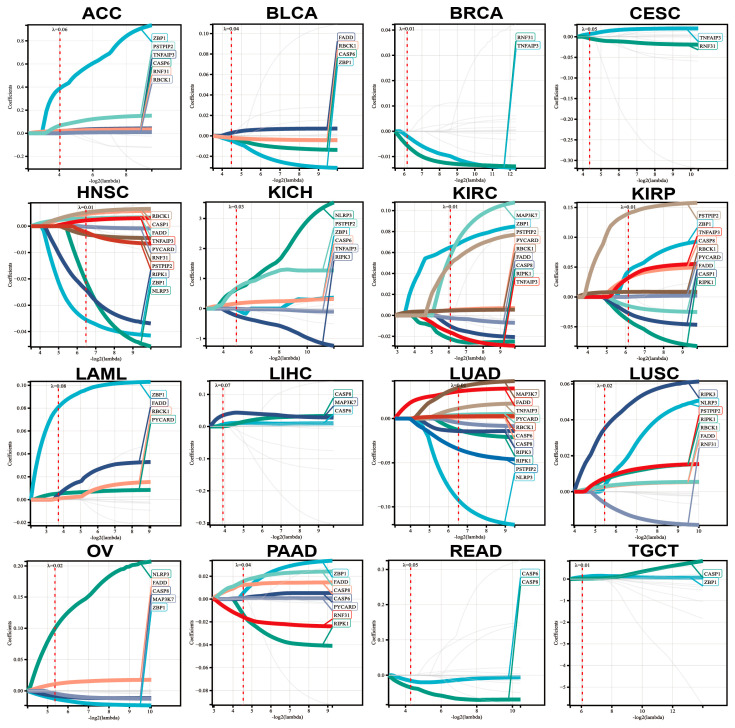
LASSO-Cox analysis on tumors to screen for meaningful genes.

**Figure 8 genes-14-01994-f008:**
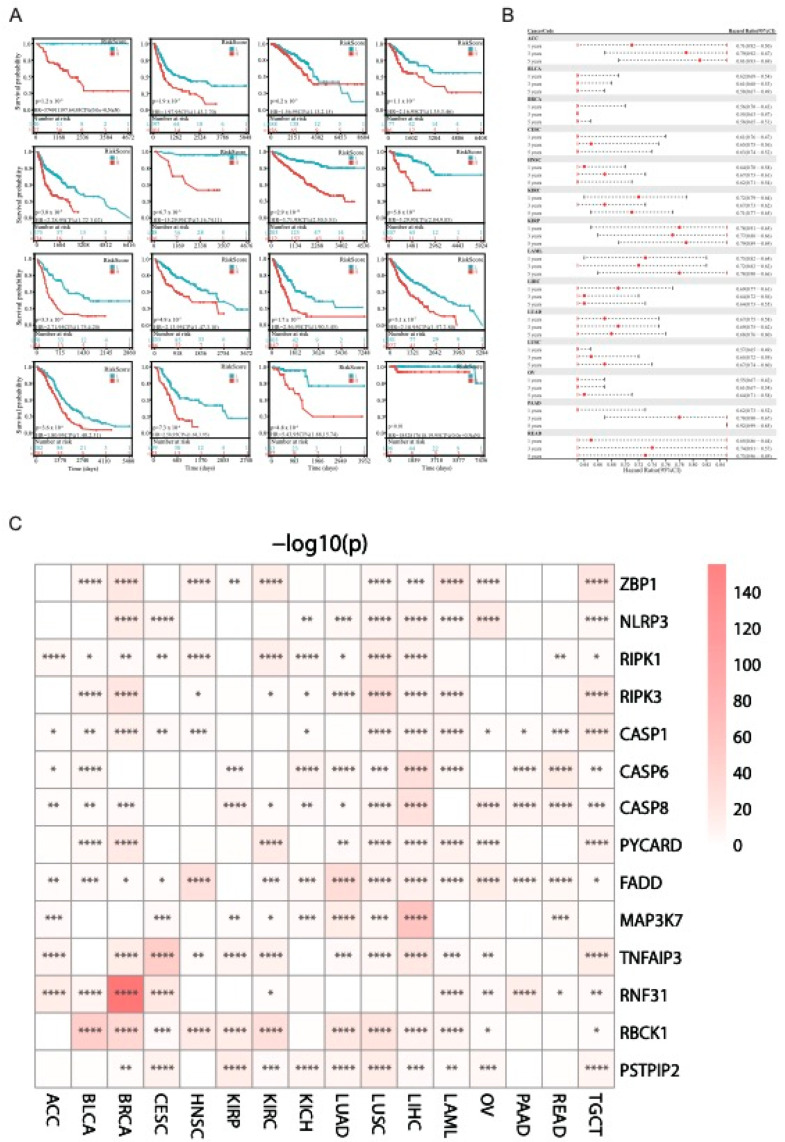
Patient survival analysis. (**A**) In different cancers, patients were divided into high and low risk score groups, and the survival differences between the two groups were compared. (**B**) Prediction of the incidence rate of different cancers after 1, 3, and 5 years. (**C**) The relationship between PANRGs and survival rates of various cancer patients. (*: *p* < 0.05, **: *p* < 0.01, ***: *p* < 0.001, ****: *p* < 0.0001).

**Figure 9 genes-14-01994-f009:**
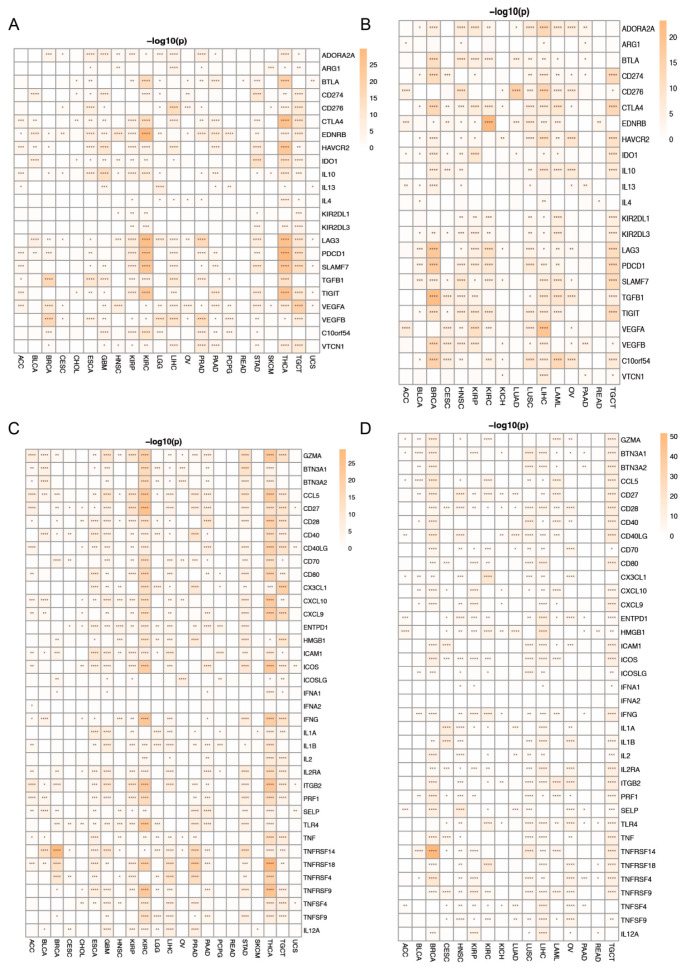
The relationship between PANRGs and immune checkpoints. (**A**) Differences between PANRGs and immune checkpoint inhibitors using clustering. (**B**) Differences between PANRGs and immune checkpoint inhibitors using LASSO. (**C**) Differences between PANRGs and immune checkpoint stimulators using clustering. (**D**) Differences between PANRGs and immune checkpoint stimulators using LASSO. (*: *p* < 0.05, **: *p* < 0.01, ***: *p* < 0.001, ****: *p* < 0.0001).

**Figure 10 genes-14-01994-f010:**
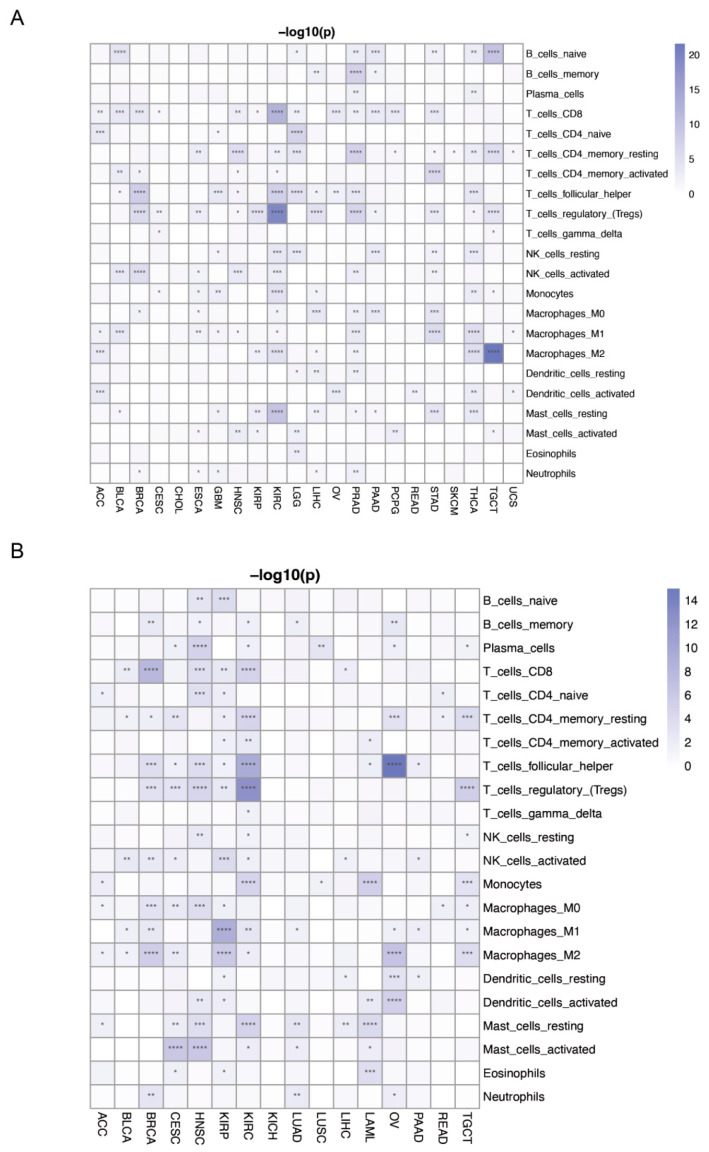
The relationship between PANRGs and immune cells. (**A**) Differences between PANRGs and immune cells using clustering. (**B**) Differences between PANRGs and immune cells using LASSO. (*: *p* < 0.05, **: *p* < 0.01, ***: *p* < 0.001, ****: *p* < 0.0001).

**Figure 11 genes-14-01994-f011:**
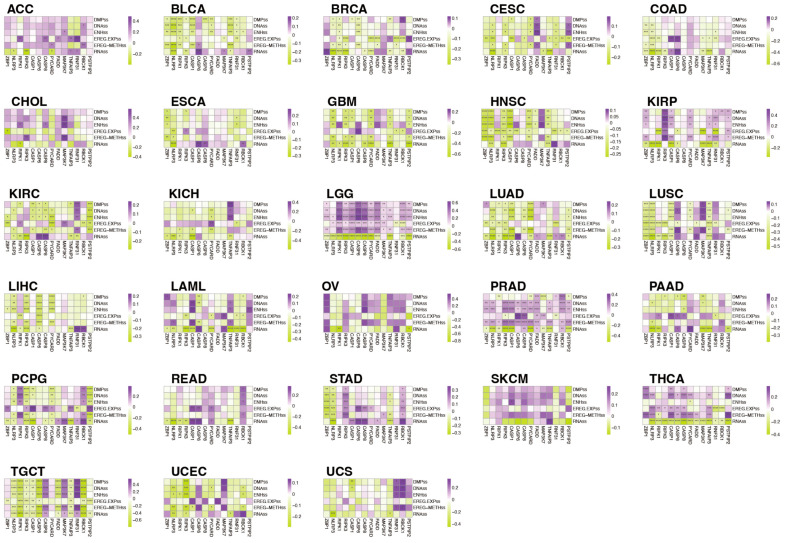
The relationship between PANRGs and tumor stemness in multiple cancers. Analysis of the relationship between 14 PANRGs and 6 tumor stem cell markers, namely, RNAs, ENHss, EREG.EXPss, DNAss, EREG-METHss, and DMPss, in 28 types of cancer. (*: *p* < 0.05, **: *p* < 0.01, ***: *p* < 0.001, ****: *p* < 0.0001).

**Figure 12 genes-14-01994-f012:**
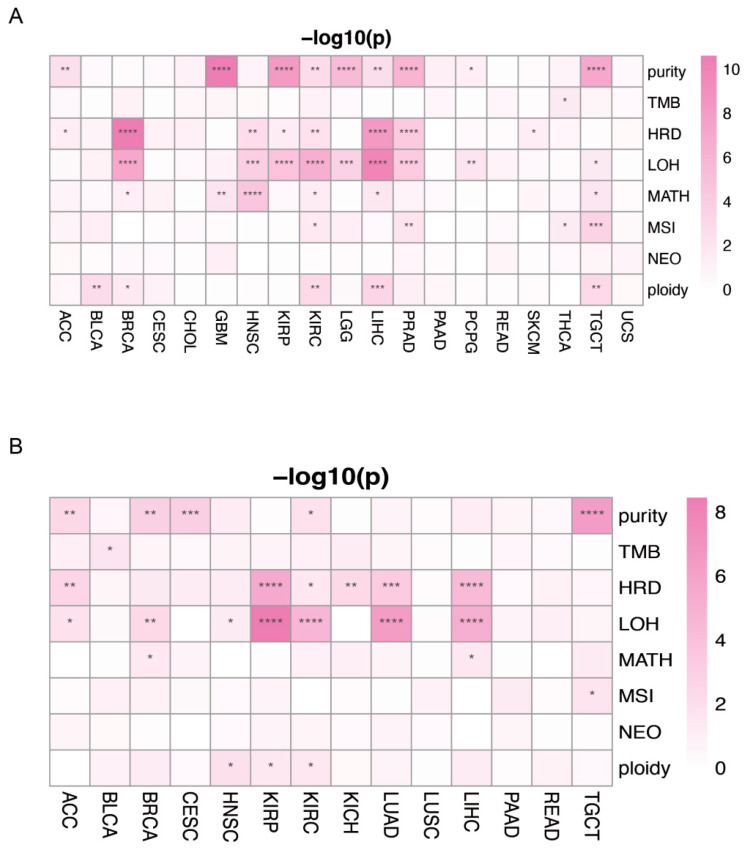
The relationship between PANRGs and tumor heterogeneity. (**A**) Differences in tumor heterogeneity among PANRGs using clustering. (**B**) Differences in tumor heterogeneity among PANRGs using LASSO. (*: *p* < 0.05, **: *p* < 0.01, ***: *p* < 0.001, ****: *p* < 0.0001).

**Figure 13 genes-14-01994-f013:**
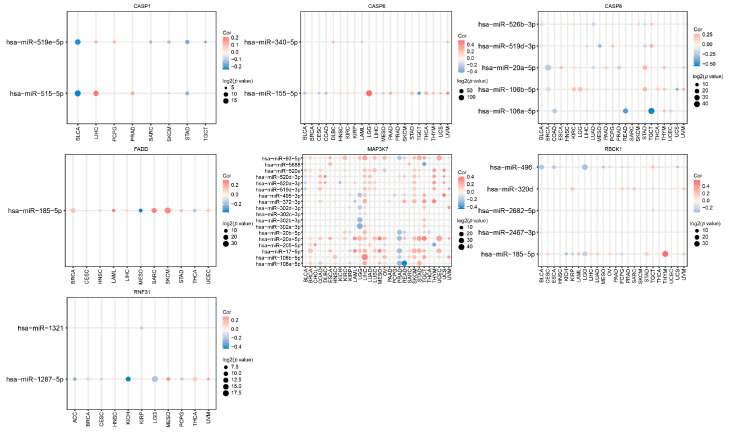
The relationship between PANRGs and miRNAs. Analysis of which tumors interact with miRNAs through PANRGs.

**Figure 14 genes-14-01994-f014:**
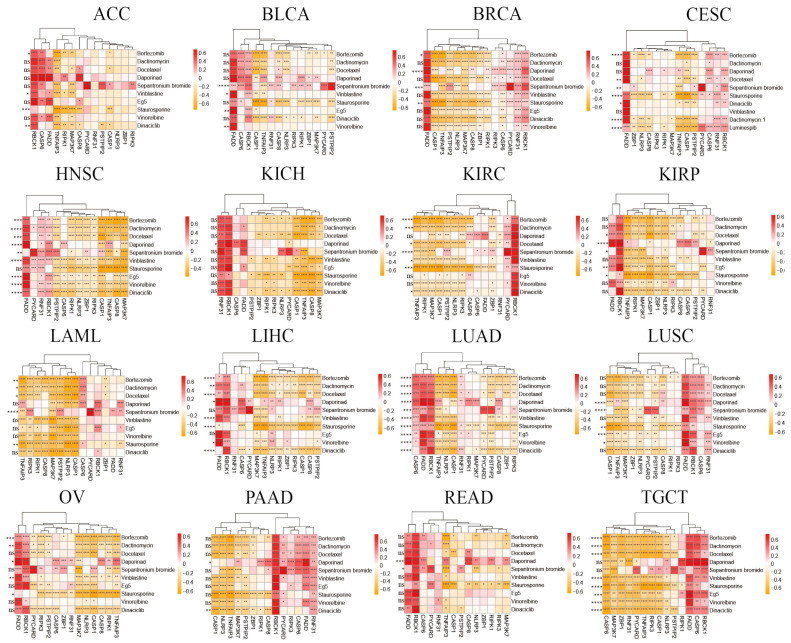
The relationship between PANRGs and drug therapy in multiple cancers. Analysis of the correlation between 10 predicted drugs and 14 PANRGs to explore potential new drugs for treating cancer. (*: *p* < 0.05, **: *p* < 0.01, ***: *p* < 0.001, ****: *p* < 0.0001).

## Data Availability

The full code used during the current study is available at https://github.com/sangmm12/PANoptosis, accessed on 16 September 2023.

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
