# Peer review of "Predicting Prognosis and Immunotherapy Response in Multiple Cancers Based on the Association of PANoptosis-Related Genes with Tumor Heterogeneity"

_genes, 2023, doi:10.3390/genes14111994_

Round 1

Reviewer 1 Report

Comments and Suggestions for Authors

The manuscript by Wang et al., presents the results of bioinformatics analyses of the expression, mutations and prognostic value of genes related to PANoptosis in different cancer types. Additionally, it shows the correlation between the expression of the PANoptosis-related genes and the infiltration of immune cells, features of steaminess, miRNA level and sensitivity to several drugs.

While the manuscript is properly constructed, it has several weak points:

1. The authors repeatedly claim that the genes associated with PANoptosis affect/influence various cancer-related phenomena. However, even the most comprehensive bioinformatics analysis does not prove a cause-and-effect relationship. It can indicate changes (e.g. in gene expression), phenomena (e.g. mutations) or correlation between some parameters, but clearly cannot prove any mechanism of action. Therefore, any suggestions regarding an impact or a mechanism should be deleted from the text (also from the title).

2. The authors analyzed the correlation of a number of genes with a number of phenomena in a number of cancers, which led to the accumulation of a lot of data, some of which is not new, and above all blurred the essence of this study. A clear indication of the novelty of this study would greatly increase the value of this paper.

3. An indication of the limitations of this study would also improve this manuscript.

4. Conclusions are too far-fetched: this study does not identify any functional roles of PANRGs. This study does not provide any evidence that any of PANRGs are crucial for tumor progression (even if there is a correlation between the expression and tumor stage, it does not mean that the gene product plays crucial role in the progression process). The last sentence in Conclusions is incomprehensible

5. It would be reasonable to explain why data for 28 cancers were analyzed. Is it the total number of cancers collected in TCGA?

Minor comments:

Some of the abbreviation (e.g. in the paragraph 2.8) are not explained.

Why does Fig. 2 consist of three separate tables?

Somehow most of the legends seem to be limited to the title, it would be reasonable to add some description.

Fig. 4: the font size used to specify the gene names and the numbers assigned to colors should be increased; in the present version it is not possible to read these names, so Fig. 4 is uninformative.  

Fig. 5C-I, Fig. 6A, Fig. 8A, Fig. 13 (names of miRNAs): The font size should be increased.

The font size in the legend to Fig. 5 should be equal.

The verb “analyze” in the legends’ titles should be replaced with a noun

The names of the genes should be written in italics.

“Simple summary” should be either deleted or completed.

Comments on the Quality of English Language

The language of the manuscript needs improvement, e.g.: Discussion, para 3: the word “dimension” seems inappropriate and the phrase “the most susceptible tumor to grade, while the most significant difference in gender is found in…” is incomprehensible in this context. These are just two examples of the phrases which are difficult to understand.

Author Response

The manuscript by Wang et al., presents the results of bioinformatics analyses of the expression, mutations and prognostic value of genes related to PANoptosis in different cancer types. Additionally, it shows the correlation between the expression of the PANoptosis-related genes and the infiltration of immune cells, features of steaminess, miRNA level and sensitivity to several drugs. While the manuscript is properly constructed, it has several weak points:Major points:1. The authors repeatedly claim that the genes associated with PANoptosis affect/influence various cancer-related phenomena. However, even the most comprehensive bioinformatics analysis does not prove a cause-and-effect relationship. It can indicate changes (e.g. in gene expression), phenomena (e.g. mutations) or correlation between some parameters, but clearly cannot prove any mechanism of action. Therefore, any suggestions regarding an impact or a mechanism should be deleted from the text (also from the title).

 Response: We thank the reviewer’s suggestion. We have removed all suggestions regarding impacts and mechanisms. (please see title, lines 253-255, 406, 418-419, 425, 429-432, 436-437, 443, 444-445, 449-450).   

2. The authors analyzed the correlation of a number of genes with a number of phenomena in a number of cancers, which led to the accumulation of a lot of data, some of which is not new, and above all blurred the essence of this study. A clear indication of the novelty of this study would greatly increase the value of this paper. 

Response: We appreciate the reviewer’s comments. We have summarized our research more clearly and re-elaborated on the novelty of the article. (please see lines 16-36, 75-83, 595-602). 

3. An indication of the limitations of this study would also improve this manuscript.

Response: We appreciate the reviewer’s comments and we clarified the limitations of this study. (please see lines 603-611). 

4. Conclusions are too far-fetched: this study does not identify any functional roles of PANRGs. This study does not provide any evidence that any of PANRGs are crucial for tumor progression (even if there is a correlation between the expression and tumor stage, it does not mean that the gene product plays crucial role in the progression process). The last sentence in Conclusions is incomprehensible.

Response: We thank the reviewer’s suggestion. We have summarized the tumors and genes that need attention more clearly, providing new directions for subsequent cancer treatment. (please see lines 595-602). Thank you for raising your question. We have made more detailed modifications to the last sentence of the conclusion. (please see lines 620-623).

 5. It would be reasonable to explain why data for 28 cancers were analyzed. Is it the total number of cancers collected in TCGA?

Response: We thank the reviewer’s suggestion. We have provided additional explanations in the corresponding parts of the text. (please see lines 96-102). Minor points:1. Some of the abbreviation (e.g. in the paragraph 2.8) are not explained.Response: We thank the reviewer’s comments and have added explanations. (please see lines 167-171). 

2. Why does Fig. 2 consist of three separate tables?

Response: We appreciate the reviewer’s comments. We have made improvements to Fig. 2. (please see line 253). 

3. Somehow most of the legends seem to be limited to the title, it would be reasonable to add some description.

Response: We thank the reviewer’s suggestion. We have added some description to the legends. (please see lines 253-255, 293-294, 403, 421-423, 439, 452-453, 472-474). 

4. Fig. 4: the font size used to specify the gene names and the numbers assigned to colors should be increased; in the present version it is not possible to read these names, so Fig. 4 is uninformative.  Fig. 5C-I, Fig. 6A, Fig. 8A, Fig. 13 (names of miRNAs): The font size should be increased.The font size in the legend to Fig. 5 should be equal.

 Response: We thank the reviewer’s suggestion and we have made changes to our figures as suggested. (please see Fig. 4, Fig. 5, Fig. 6, Fig. 8, Fig. 13).

 5. The verb “analyze” in the legends’ titles should be replaced with a noun

Response: We appreciate the reviewer’s comments. We have changed all "analyze" in the legends’ titles to "analysis". (please see lines 253, 277). 

6. The names of the genes should be written in italics.

Response: We thank the reviewer’s suggestion. We have changed all the genes to italics. 

7. “Simple summary” should be either deleted or completed.

Response: We thank the reviewer’s suggestion and we have deleted “Simple summary”.

 Comments on the Quality of English LanguageThe language of the manuscript needs improvement, e.g.: Discussion, para 3: the word “dimension” seems inappropriate and the phrase “the most susceptible tumor to grade, while the most significant difference in gender is found in…” is incomprehensible in this context. These are just two examples of the phrases which are difficult to understand.

Response: We thank the reviewer’s comments. We have carefully revised the descriptions that are not accurate in the whole manuscript. In addition, the possible grammatical and spelling errors have also been further edited by an English editor of MDPI.

Reviewer 2 Report

Comments and Suggestions for Authors

In this study by Wang et. al., the authors proposed PANoptosis related genes as potential biomarkers that help predicting tumor prognosis and responses to immunotherapy in Pancancer. Overall, the study is well written. The authors performed a detailed and extensive analysis to reach a conclusion. However, my only concern is the conclusion of their findings is not clear. Rather than providing general conclusion, the authors need to be specific based on their findings.

Author Response

In this study by Wang et. al., the authors proposed PANoptosis related genes as potential biomarkers that help predicting tumor prognosis and responses to immunotherapy in Pancancer. Overall, the study is well written. The authors performed a detailed and extensive analysis to reach a conclusion. However, my only concern is the conclusion of their findings is not clear. Rather than providing general conclusion, the authors need to be specific based on their findings.

Response: We thank the reviewer’s suggestions and we have provided a clearer summary of the research results. (please see lines 595-602).

Round 2

Reviewer 1 Report

Comments and Suggestions for Authors

The manuscript has been thoroughly revised. All the concerns have been taken into account. Thank you for your effort.